# Spatiotemporal changes in heat stress exposure in India, 1981-2023

Arpit Shah [1,2], Anish Sugathan[3,4], Deepak Malghan [1,5], Rockli Kim[6,7] & S. V. Subramanian [8,9] ✉

India is amongst the most susceptible regions in the world to extreme heat stress because of climate change. In this study, we present a comprehensive analysis of the spatiotemporal evolution of heat stress exposure (HSE) across India's districts, which serve as the primary administrative units below the State level. Our analysis covers a long temporal duration (1981-2023) and uses data with high spatial (0.1°) and temporal granularity (hourly). By focusing on the district, we ensure that our findings are relevant for policymakers. We find a 3.3% increase in the average HSE duration across districts when comparing 1981-1995 and 2011-2023. We provide evidence for variation in the evolution of HSE by time of day, time of year, and across regions in India. We also provide estimates of changes in outdoor occupational exposure from 2019-2023. Our research highlights the importance of considering HSE in addition to average temperature changes and has important implications for public health practitioners and policymakers.

India is projected to witness an intensification of heat wave patterns combined with increasing human mortality under current warming trends[1-3]. Excess heat has health implications, ranging from mortality and morbidity to adverse birth and mental health outcomes[4]. Heat can influence health indirectly through its influence on agricultural production and food availability[5]. High heat stress also has macro-economic implications, as it can reduce physical work capacity and enhance occupational risks[6-8]. The International Labor Organization estimates suggest that 1 billion workers worldwide are exposed to high heat episodes[9].

Outdoor workers are particularly vulnerable to increasing heat stress, and South Asia is projected to be the most vulnerable region on this front, with its workforce predominantly employed in outdoor occupations such as agriculture and construction[9]. The impacts of occupational heat stress can be further exacerbated in regions with high overnight temperatures, as adequate rest can become challenging with prolonged exposure[10]. Occupational heat stress is also modulated by inter-group inequalities, with recent evidence from India showing that workers from marginalized caste groups are more exposed to outdoor occupational heat stress[11].

Prior work on understanding the historical evolution of heat stress in India has typically relied on datasets that provide daily temperature information with a spatial resolution of 0.25°−0.5°[1,12,13]. Recent state-of-the-art datasets such as ERA-5 Heat make it possible to avail hourly data at a spatial resolution of 0.25°[14], while the HiTiSEA dataset provides daily data at a higher spatial resolution of 0.1°[15]. The HiGTS dataset, published recently, provides high spatial (0.1°) and temporal (hourly) resolution for the years 2000 to 2023 globally[16]. Currently, no datasets provide spatiotemporal thermal information at a high resolution over India and South Asia for a longer time period. In addition, most datasets rely on average or instantaneous temperature values to assess thermal comfort. However, research indicates that the duration of exposure to stressful heat conditions is also an important factor influencing human health[17]. A higher duration of exposure is more

[1]Centre for Public Policy, Indian Institute of Management Bangalore, Bengaluru, India. [2]Faculty of Arts and Sciences, Harvard University, Cambridge, MA, USA. [3]Strategy, Indian Institute of Management Ahmedabad, Ahmedabad, India. [4]Centre for Sustainability and Corporate Governance, Indian Institute of Management Ahmedabad, Ahmedabad, India. [5]Asia Centre, Stockholm Environment Institute, Bangkok, Thailand. [6]Division of Health Policy and Management, College of Health Science, Korea University, Seoul, South Korea. [7]Interdisciplinary Program in Precision Public Health, Department of Public Health Sciences, Graduate School of Korea University, Seoul, South Korea. [8]Harvard Center for Population and Development Studies, Cambridge, MA, USA. [9]Department of Social and Behavioral Sciences, Harvard T.H. Chan School of Public Health, Boston, MA, USA. ✉e-mail: svsubram@hsph.harvard.edu

likely to produce physiological strain in the human body, leading to health issues because of heat[18].

In this research, we analyze the long-term evolution of heat stress exposure in India. Compared to existing datasets such as HiTiSEA and HiGHTS, we analyze a longer temporal span (1981 to 2023), and leverage data at a higher spatiotemporal resolution (0.1°, hourly). By focusing on India's districts as the unit of analysis, our work has relevance for governance and policymaking. Our choice of district is based on its role in India's administrative setup. The district is an important unit of local governance in India, and the district administration plays a key role in disaster preparedness and response efforts[19]. District administrations are responsible for directing and implementing government programs in India, and the country's public health outcomes are directly dependent on the performance of districts in delivering developmental goals[20]. For instance, the Aspirational Districts program of the Government of India, which aims to transform India's most underdeveloped districts, underscores the importance of the district in policy implementation[21]. In particular, the district is one of the primary administrative levels at which Heat Action Plans are prepared in India[22].

For our analysis, we use the Universal Thermal Climate Index (UTCI), a multivariate parameter that characterizes heat exchanges between the human body and the environment[23]. Our choice of the UTCI as a measure of heat stress is based on several reasons. First, the UTCI is a reliable alternative to the Wet Bulb Globe Temperature, which is considered the best measure for understanding the human physiological response to heat[23]. Recent work shows that the UTCI performs as well as, or in some cases better than, the WBGT in predicting body temperature, human thermal perception and labor loss in a range of climatic conditions[24]. Second, the UTCI incorporates the impact of air temperature, relative humidity, wind speed, and solar radiation, and is particularly appropriate in understanding heat stress faced by outdoor occupational workers[23,25]. Third, the reliability of the UTCI in assessing thermal comfort has been validated across a range of climatic conditions[26].

In this work, we analyze the evolution of UTCI levels in India's districts from 1981 to 2023. We find an increase in average UTCI levels in India across regions. Daily minimum UTCI levels have risen faster than daily maximum UTCI levels, leading to a narrowing of the diurnal UTCI range over time. We propose a metric called Heat Stress Exposure (HSE), which occurs when UTCI crosses specific threshold values. Our rationale is that increases in average UTCI alone may not result in heat stress if the thermal metric does not cross certain thresholds that are considered stressful for human health. Our study reveals that HSE in India has risen over time when measured by the number of hours per day that the UTCI exceeds 32 °C—a threshold that is considered stressful, especially if individuals are outdoors doing work requiring physical exertion[11,23]. We match data from the Consumer Pyramids Household Survey (CPHS) with UTCI information to provide estimates for how outdoor occupational heat stress exposure has evolved in India from 2019 to 2023. Our analysis also highlights caste-based inequality in outdoor occupational heat exposure. Overall, our work makes an important contribution by presenting a long-term, policy-relevant analysis of trends in thermal stress in India using high-resolution spatiotemporal data.

## Results
### Comparing UTCI and HSE
Figure 1 compares the average UTCI and HSE across India at a spatial resolution of 0.1° for 1981 and 2023 (refer Supplementary Fig. 1 for country-level UTCI calculations). We find that HSE has increased faster than the average UTCI in this period. While the average UTCI has changed from 22.5 °C in 1981 to 23.4 °C in 2023 (3.6% increase), the average daily HSE has changed from 5.1 h/day to 5.6 h/day (10.8% increase). Panels (c) and (f) illustrate our point regarding the importance of considering both average UTCI and HSE together. Panel (c) shows that

UTCI has converged across pixels—the negative slope indicates that pixels that were hotter in 1981 have seen smaller increases in UTCI between 1981 and 2023. In contrast, the positive slope for the regression line in Panel (f) shows that HSE has diverged across pixels.

### Trends in HSE over time
Figure 2 presents changes in HSE over time in India. For easier interpretation, we consider three time periods (1981 to 1995, 1996 to 2010, and 2011 to 2023). Panel (a) presents changes in HSE by time of day across the three time periods. HSE predominantly occurs between 8 am and 6 pm, when UTCI values are at their highest. We observe increased HSE levels across daytime hours (especially between 8 am and 6 pm). Overall, the average daily HSE in the 2011–2023 period (335 min/day across India) is 3.3% higher than the HSE in the 1981–1995 period. Panel (b) presents similar results for different months of the year. Except for a slight dip in June and July during the 1996–2010 period, we again observe a secular increase in HSE over time for other months. For visual clarity, Panels (c) and (d) present the difference between the respective curves in Panels (a) and (b), respectively, with the 1981–1995 period serving as a reference. Panel (e) presents the distribution of average daily HSE levels at the district level. As we move ahead, we find that the curve shifts towards the right - a larger number of districts experience HSE levels at the higher end of the scale. Panel (f) presents the robust trend estimates of HSE between 1981 and 2023 at the district level, plotted against the average HSE of the district in 1981. Each point represents a district, color-coded into different regions based on geographical location in India. The line shows the OLS regression for the set of points (722 districts) in the graph. The positive slope implies a divergence in HSE levels across districts. HSE levels have increased more in districts that already had higher HSE levels. We also observe a clustering of values at the regional level. Average HSE has increased the least in South India (blue dots) at an average of 0.08 h/day, with many districts also witnessing a decreasing trend in HSE levels. In contrast, the Central (orange dots) and Eastern (greenish-yellow dots) regions show an increase of 0.45 h/day and 0.46 h/day, respectively. The results highlight inter-regional variations in HSE trends across the country.

### Trends in UTCI over time
Panel (a) of Fig. 3 presents changes in India's daily maximum, median, and minimum UTCI values over time. Each point in the graph represents the annual mean of daily values of the respective parameter for the given year. While all UTCI metrics show an increase, a visual comparison of the three panels indicates that daily median and minimum UTCI values (the central and right panels, respectively) have risen faster than the daily maximum UTCI values (left panel). Daily maximum UTCI has risen at 0.26 °C over this period, while median and minimum UTCI values have increased by 0.96 °C and 1.05 °C, respectively. Panel (b) presents the resultant shrinking of the diurnal UTCI range at the district level (based on robust Sen's slope-based trend estimates). We find that the diurnal range has shrunk across most districts, with the exception of a few districts in Eastern India. The shrinking of the diurnal range impacts public health and is associated with cardiovascular diseases, mortality, and morbidity[27].

### Trends in HSE by quarter
Figure 4 compares HSE levels across time for different quarters of the year. We observe variations in how exposure levels have evolved across quarters and regions. For instance, exposure increases in the Northern Plains in Quarter 3 in Panel (c), while increases are more prominent in Southern India in Quarter 4 in Panel (d). Average exposure across the country increases by 6.5%, 2.6%, and 9.9% in Quarters 1, 3, and 4, respectively, as seen in Panel (e). In contrast, average exposure does not increase meaningfully in Quarter 2. The lack of increase

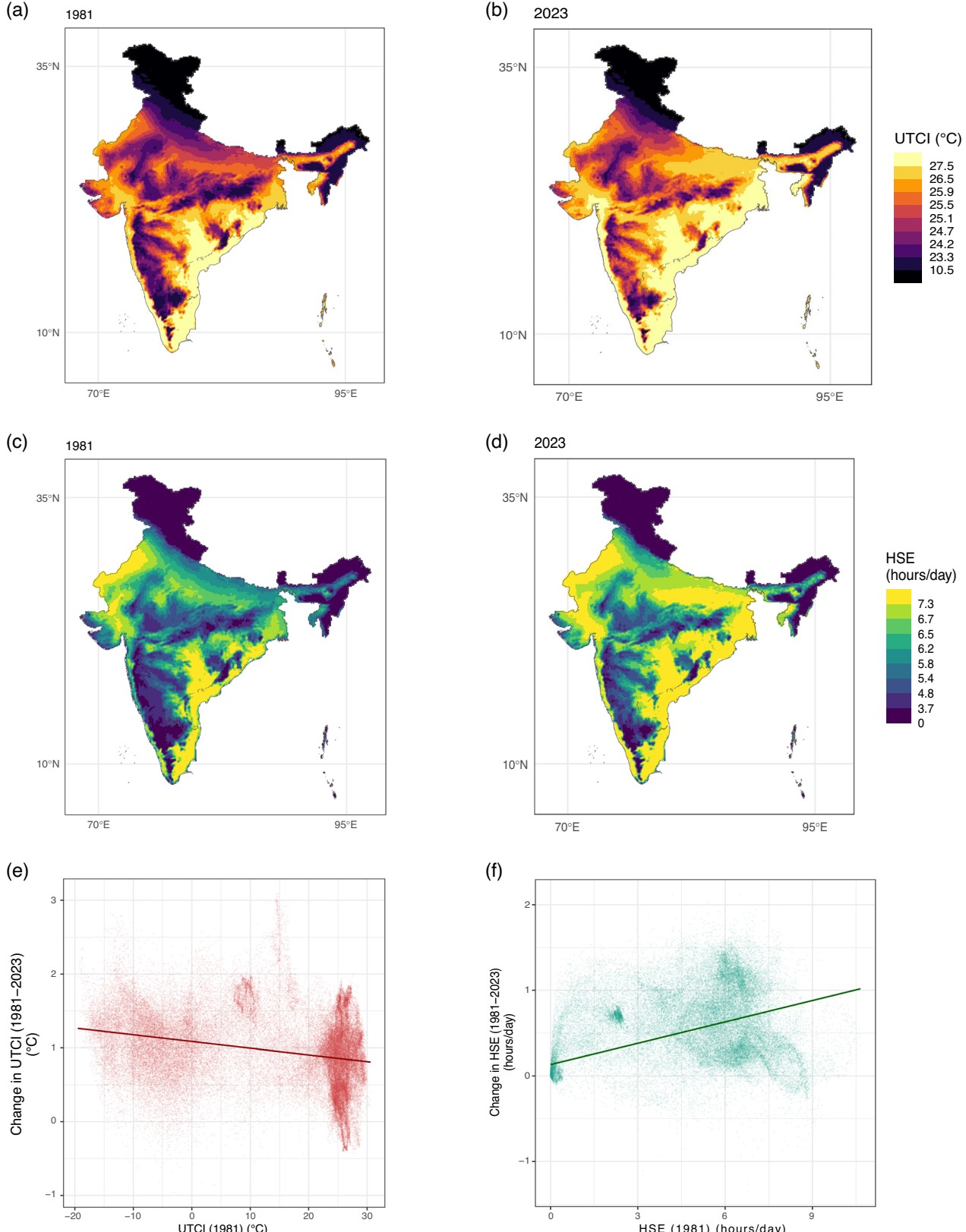

**Fig. 1 | Comparison of Universal Thermal Climate Index (UTCI) and Heat Stress Exposure (HSE) in India for 1981 and 2023. a** Average UTCI (°C) values for different parts of India at a spatial resolution of 0.1° for 1981. **b** Same as (**a**) for the year 2023. **c** Average UTCI in 1981 for each pixel (0.1° resolution) plotted against the change in UTCI for the pixel between 1981 and 2023. Note: We refer to a ERA5 Land grid point as a pixel in this manuscript. **d** Average daily heat exposure (HSE) (hours/ day) for different parts of India at a spatial resolution of 0.1° for 1981. **e** Same as (**d**) for the year 2023. **f** Average HSE in 1981 for each pixel (0.1° resolution) plotted against the change in HSE for the pixel between 1981 and 2023. The lines in (**c**) and (**f**) represent the OLS regression line for the set of points. HSE occurs when UTCI exceeds 32 °C in a given hour. The values for (**d**–**f**) are derived by averaging the mean number of hours for which UTCI exceeds 32 °C for each day of the year.

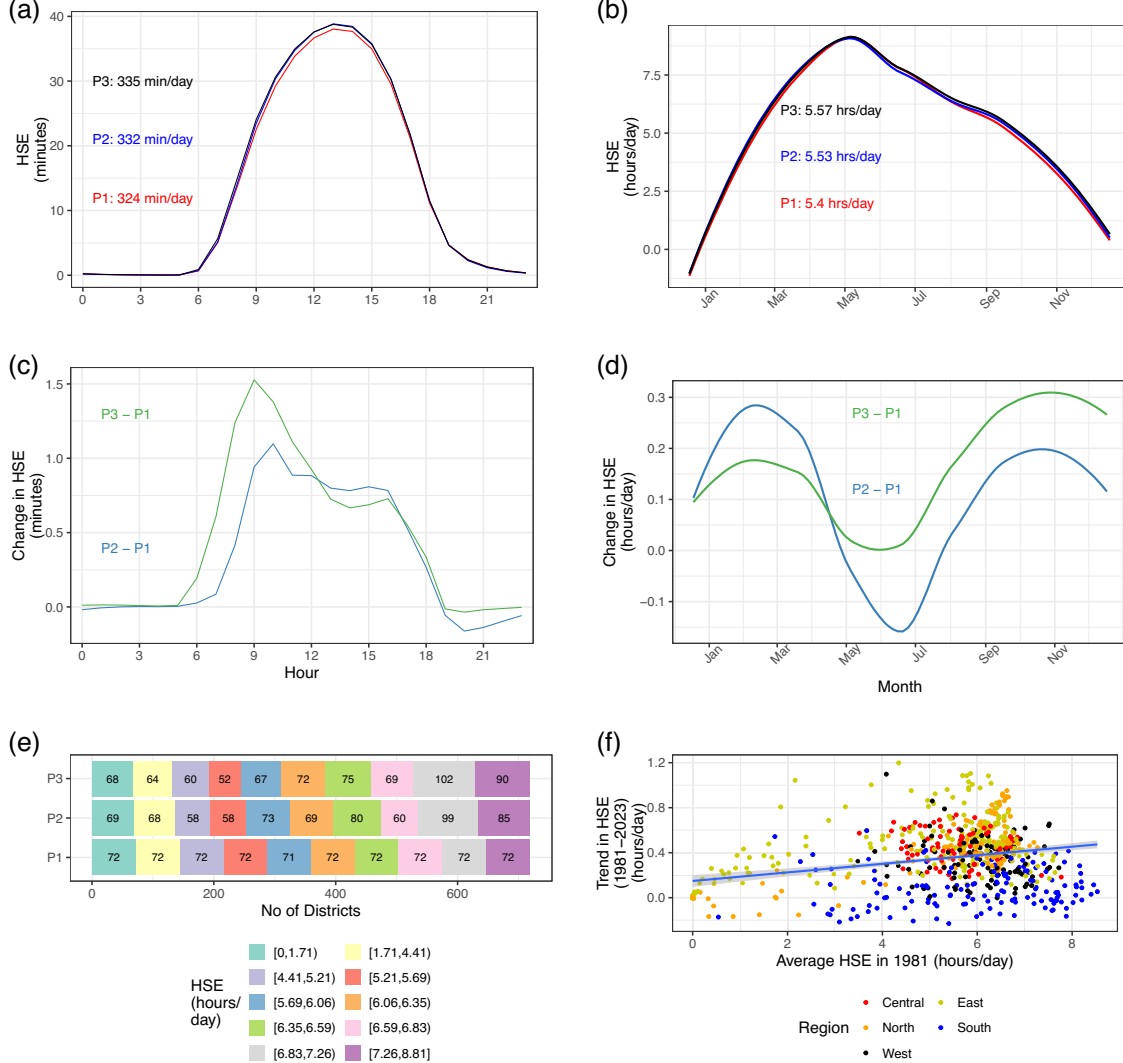

**Fig. 2 | Change in Heat Stress Exposure (HSE) in India (1981 to 2023). a** Average HSE at different times of day for three time periods. The text in the panel is the average daily exposure during the time period (area under the curve). P1: 1981–1995, P2: 1996–2010, P3: 2011–2023 (**b**) Average daily HSE in different months of the year for three time periods. The text in the panel is the average daily exposure during the time period. **c** This panel represents the change in HSE between the defined time period (as compared to the 1981–1995 period, which serves as a reference) for different times of day. The values represent the difference between the respective curves in (**a**). **d** This panel represents the change in HSE between the defined time period (as compared to the 1981–1995 period, which serves as a reference) for different times of the year. The values represent the difference between the respective curves in (**b**). **e** Histogram showing the distribution of average daily HSE at the district

level for three time periods. District UTCI values are calculated as a weighted average of pixel UTCI values (0.1° resolution) and the areal overlap between the pixel and the district. HSE occurs in a district when the district UTCI exceeds 32 °C in a given hour. **f** The panel plots the average HSE for each district in 1981 on the X-axis. For each district, this value is the mean of the daily HSE levels for the year. The Y-axis plots the change in HSE between 1981 and 2023. The Y-axis is based on the robust Sen's slope of the detrended time-series of HSE level for each district for the 1981-2023 time period (details in Methods). Each point represents a district (a total of 722 districts). The colors represent regions in India. Each district is coded into a separate region based on its location in the country (details provided in Supplementary Table 1). The line is the OLS regression for the set of points in the graph. The shaded region represents the 95% confidence interval for the regression line.

in Quarter 2 might also be a high-base effect. At 8.94 h/day in the 1981-1995 period, exposure levels in the April- June quarter were already high in the initial period.

### Trends in HSE by time of day
Figure 5 compares exposure levels by day and night for three time periods. We find that average exposure levels during the daytime (6 am to 6 pm) increase by 3.3%, while HSE levels at night remain stable over time. A visual inspection of Panel (a) indicates that HSE is increasing in the northern region of India. These results highlight the importance of assessing both average temperature patterns and HSE levels simultaneously, as the relationship between the two is not linear. While Fig. 3 shows that night-time temperatures are increasing faster than daytime

temperatures on average, we find that HSE increases during the daytime and not at night. This is primarily because night-time UTCI remains largely lower than the threshold value of 32 °C.

### Trends in continuous exposure events
Figure 6 compares HSE exposure events for three time periods. We define an HSE exposure event as a continuous period of time during which UTCI exceeds 32 °C. Panels (a), (b) and (c) show that the frequency of HSE exposure events has increased (from 276 to 280), the interval between HSE exposure events has declined (from 21.7 h to 21.3 h), and the duration of the average exposure event has increased (from 7.96 h to 8.05 h). This implies that heat episodes are getting longer with reduced time to rest between episodes.

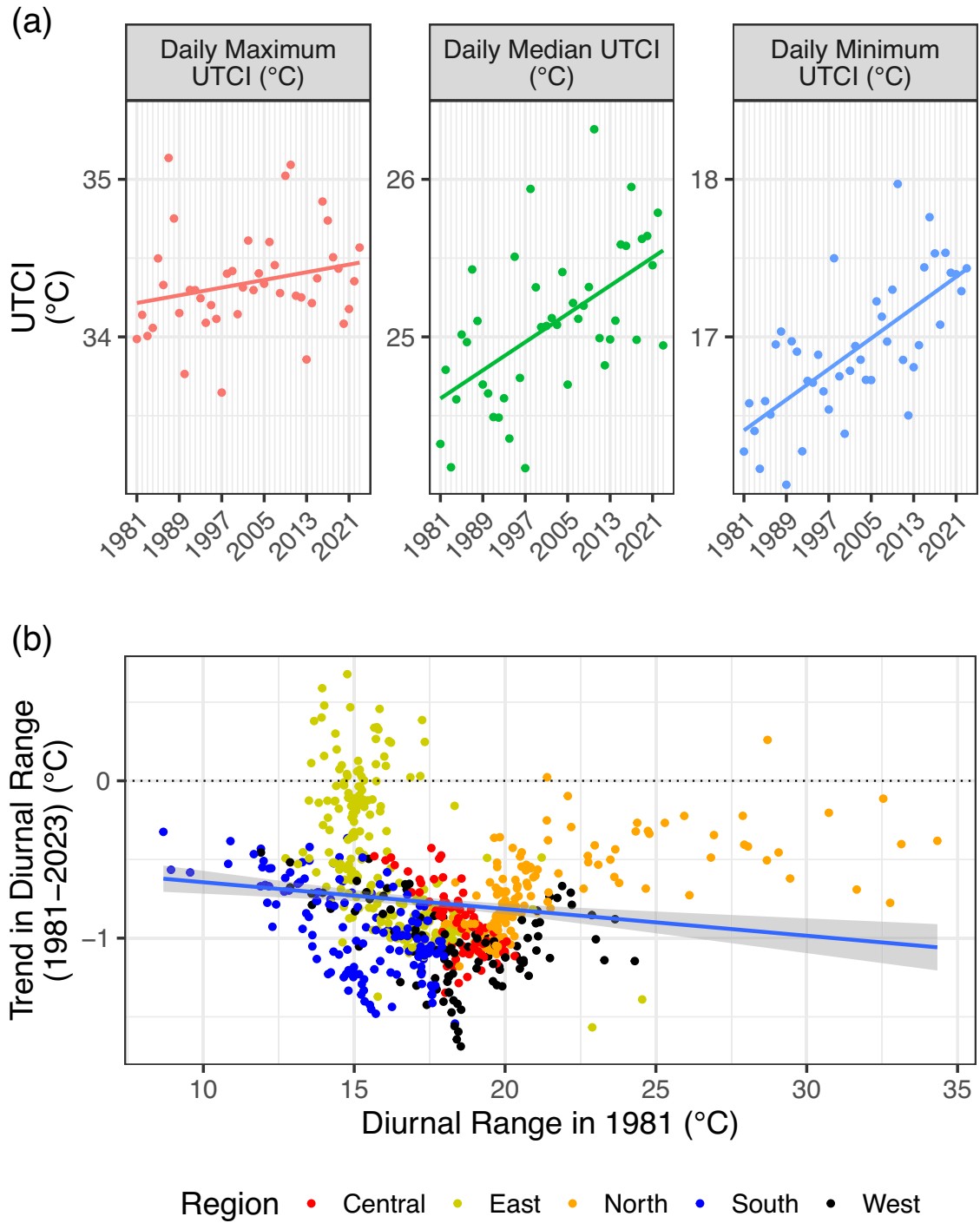

**Fig. 3 | Trends in the minimum, median and maximum daily Universal Thermal Climate Index (UTCI) and the impact on the diurnal range. a** The left panel presents the change in the daily maximum UTCI over time in India, from 1981 to 2023. Each point represents the mean of the daily maximum UTCI values for a given year. The line represents the results of a simple OLS model fitting the points. The central panel presents the change in the daily median UTCI from 1981 to 2023. The right panel presents the change in the daily minimum UTCI from 1981 to 2023. Points and lines in the central and right panels have similar interpretations as the left panel. The three panels have the same increments on the Y-axis (with different ranges). **b** The panel presents the trend in diurnal UTCI range (based on robust Sen's slope estimates) at the district level plotted against the average diurnal UTCI range for the district in 1981. Each point represents a district. The line is the OLS regression for the set of points in the graph. The shaded region represents the 95% confidence interval for the regression line.

## HSE in outdoor occupations

Figure 7 matches occupational data from India's Consumer Pyramids Household Surveys (CPHS) with UTCI information to present outdoor occupational exposure for workers from September 2019 to December 2023. The surveys occur thrice a year and cover the months indicated in the graph. Outdoor occupational exposure occurs when an individual is working outdoors at a time when the UTCI exceeds 32 °C. Panel (a) shows that the aggregate exposure and the proportion of outdoor work hours that are under exposed have increased from 2019 to 2023 (it is important to note that the lower values for 2020 and the first half of 2021 might be a result of pandemic-related restrictions). Panel (b) presents inequality in exposure across India's caste groups.

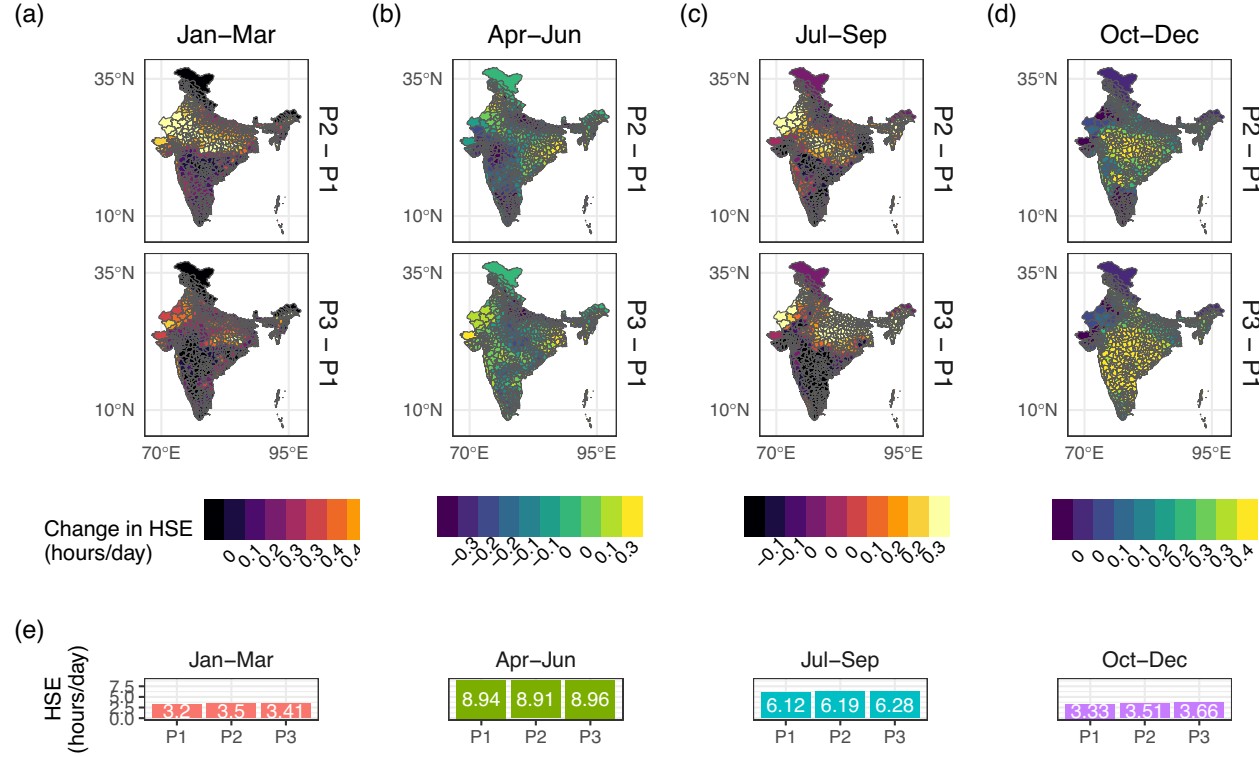

**Fig. 4 | Change in Heat Stress Exposure (HSE) for different quarters of the year.** The figure presents changes in HSE levels for 1996–2010 and 2011–2023 (compared to the HSE levels in 1981–1995) for different quarters of the year (Panels (a–d)). **e** presents the average HSE across all districts for the same time periods for different quarters of the year.

Marginalized caste groups in India, which include the Scheduled Castes (SC) and Scheduled Tribes (ST) have greater outdoor occupational exposure than the dominant groups which include the Other Backward Classes (OBC) and the residual Others.

## DISCUSSION

Scientists have argued that increases in moist heat stress in India can be driven both by global warming[28] and by the widespread adoption of irrigation in the country[13]. In combination, these factors make India one of the world's most vulnerable regions to excess heat. Although official government statistics report low heat-related mortality, experts estimate that the recent heat wave in 2024 had a death toll in the thousands[29]. Group-based disparities exacerbate the problem, with vulnerable groups most likely to be working outdoors in conditions that force them to bear heat stress[11]. In this context, it is important for policymakers to develop a better understanding of trends in heat exposure across the country.

In this paper, we present evidence for increasing thermal stress in India from 1981 to 2023. Our approach provides insights at the policy-relevant scale of the district over a long temporal duration (1981 to 2023) using data with a high spatial (0.1°) and temporal resolution (hourly). We provide a nuanced characterization of how thermal stress has evolved across the country over time. Our analysis maps both the intra-day and intra-year evolution of thermal stress across different regions in India. We also provide estimates of changes in outdoor occupational heat stress exposure.

Our focus on exposure to stressful UTCI that crosses a certain threshold is informative from a public health perspective. While prior research has focused on extreme heat wave events[1,30], it is also important to characterize heat stress that individuals might get

exposed to on a daily basis, especially when they are working outdoors. Our analysis shows an increase in daily outdoor UTCI exposure for most regions in India, when measured as the number of hours per day that UTCI crosses a threshold value of 32 °C. While the highest HSE occurs during the April-June quarter of the year, which is the hottest season in India, we find evidence for increases in exposure over time in other quarters of the year. Our findings imply that policymakers must be attentive to heat stress throughout the year, not just during the summer period when heat waves are more common.

We also contribute to the ongoing debate on how the diurnal temperature range has changed over time. Our results highlight the difference between using air temperature values and alternate indicators such as UTCI. While recent work has not agreed on how the global diurnal temperature range based on air temperature has evolved in the last four decades[31,32], we find that the diurnal range based on UTCI measurements has shrunk in India in the same period. Minimum daily UTCI values may be rising faster than maximum daily temperature because of changes in factors such as night-time humidity, which are part of the UTCI metric but are not accounted for in air temperature measurements[23,25]. Future research should examine different components of the UTCI to understand the drivers behind our findings in this work.

While we focus primarily on Indian districts as the unit of analysis for this paper, it is important to note that our methodology can be used to map changes in heat stress at finer geographical scales. At a spatial resolution of 0.1°, we have an average of 45 hourly UTCI readings per district.

We would also like to acknowledge the limitations in our approach. Since the UTCI is computed from hourly meteorological

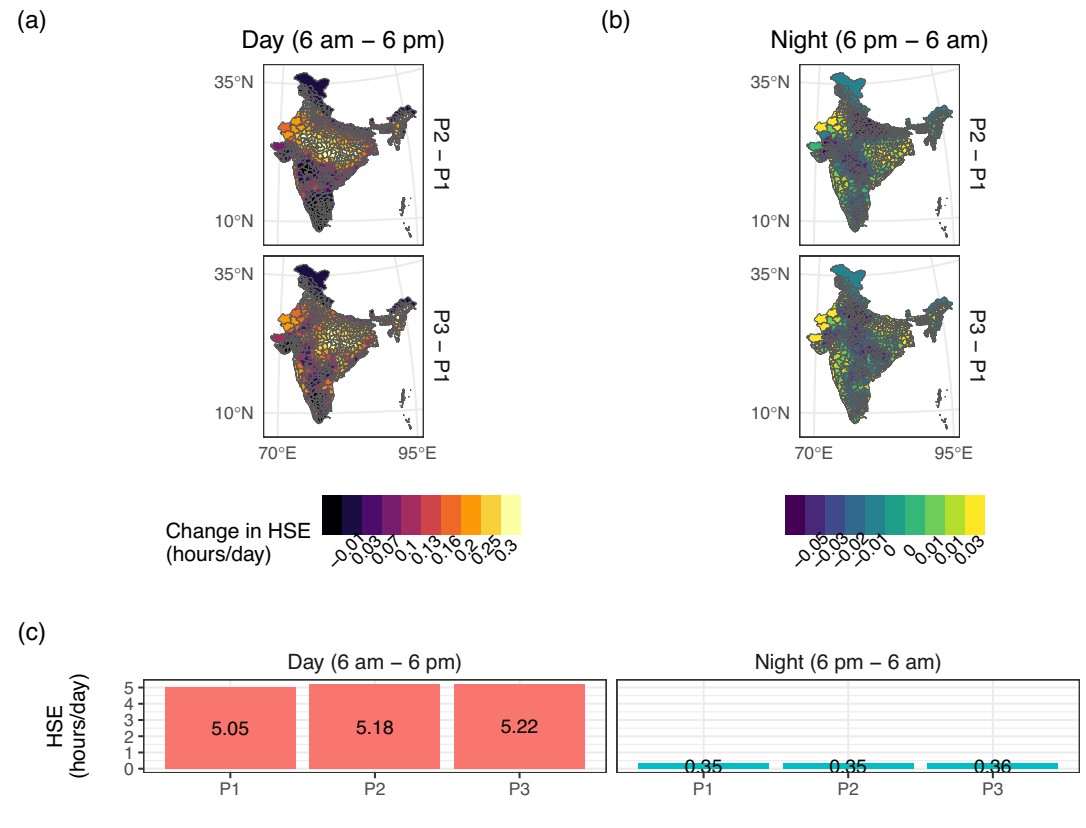

**Fig. 5 | Change in Heat Stress Exposure (HSE) for different times of day.** The figure presents changes in HSE levels for 1996–2010 and 2011–2023 (as compared to the HSE levels in 1981–1995) for different times of day in Panels (**a**) and (**b**). **c** presents the average exposure across all districts for the same time periods for different times of day.

variables available from the ERA5 and ERA5-Land datasets, biases that exist in the original data are also present in our computation. A complete discussion of potential biases in UTCI calculations is beyond the scope of this paper and is available elsewhere[15].

Overall, our analysis provides a complete characterization of the evolution of the UTCI and heat stress exposure in India, accounting for both average UTCI levels and exposure when UTCI crosses certain thresholds. While we use 32 °C as our threshold value, it is important to note that the human response to heat exposure depends on the nature of the activity being performed—lower thresholds may lead to heat strain for activities involving greater exertion. From the perspective of occupational safety for outdoor workers, it would be important for future research to calibrate UTCI thresholds specific to the nature of the activity being performed and evaluate exposure accordingly.

## Methods
### UTCI calculations
All variables used to compute the UTCI were retrieved from the ERA5 and ERA5-Land datasets. The variables used include wind speed, dewpoint temperature, air temperature, surface pressure, thermal radiation fluxes, and solar radiation fluxes (refer Supplementary Table 2 for details). The ERA5 dataset provides hourly data on all variables with a spatial resolution of 0.25°[33]. The ERA5-Land dataset leverages the land component of the ERA5 data to provide hourly data with an improved spatial resolution of 0.1°[34]. The ERA5 and ERA5-Land datasets have the finest spatial and temporal resolution among global reanalysis datasets available currently[16,35–37].

Using these variables, we calculated the UTCI following the approach used by Yan et al. (2021)[15]. The UTCI is defined as an

equivalent ambient temperature for a reference environment that would produce the same physiological response in a human body exposed to the actual environment[25]. The UTCI assumes behavioral changes in clothing insulation in response to the actual environment[38]. The reference environment is defined as weather conditions with calm air (10-m wind speed of 0.5 m/s), mean radiant temperature equal to air temperature, 50% relative humidity, water vapor pressure of 20 hPA, and a human performing an activity that involves a metabolic rate of 135 W/m². The calculations use a sixth-order polynomial regression function to calculate the UTCI[39], which, in simpler form, is provided in Eq. (1). Eq. (2) provides the detailed form.

$$UTCI = T_a + f\left(T_a, v_a, e, T_{mrt} - T_a\right) \quad (1)$$

$$UTCI = T_a + \sum_{\substack{i,j,k,l > 0 \\ i+j+k+l \le 6}} c_{ijkl} T_a^i \left(T_{mrt} - T_a\right)^j v_a^k e^l \quad (2)$$

where $T_a$ is the 2 m air temperature, $v_a$ is the 10 m wind speed, $e$ is the water vapor pressure, $T_{mrt}$ is the mean radiant temperature. The approach provided by Yan et al. (2021)[15], which we adapt, uses coefficient values for $c_{ijkl}$ from the work done by Brode et al. (2011)[39]. We calculate $T_{mrt}$ and $e$ following the details in the HiTiSea dataset[15]. Details of the variables required for the computation are provided in Supplementary Table 2. Supplementary Fig. 2 provides a comparison of the UTCI and the Wet Bulb Globe Temperature (WBGT) for India's districts from 2020–22. We conduct this analysis to highlight the reliability of the UTCI as a proxy for the WBGT.

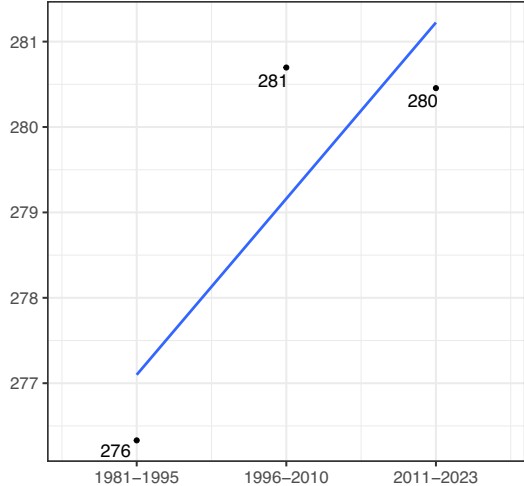

(a) Number of continuous exposure events per year

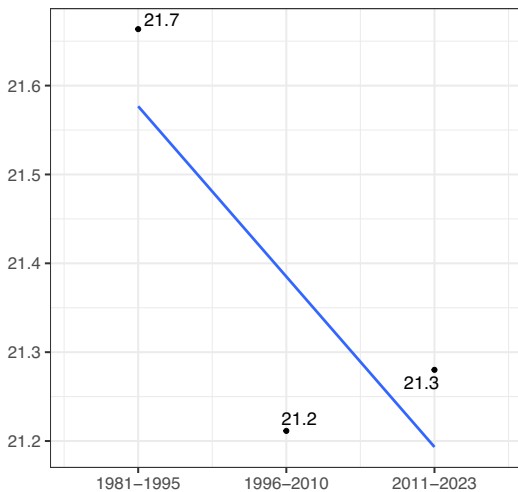

(b) Interval between consecutive exposures (hours)

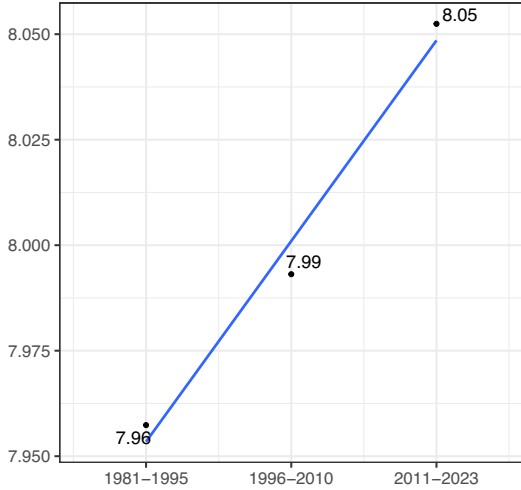

(c) Duration of exposure events (hours)

**Fig. 6 | Change in Heat Stress Exposure (HSE) exposure events.** The figure presents changes in HSE exposure events. An exposure event is defined as a continuous period of time during which UTCI exceeds 32 °C. The numbers represent the average across India's districts for the specified time period. Panels (**a**) presents the average number of exposure events in a district over time. Panel (**b**) presents the average interval between consecutive exposure events in a district over time. Panel (**c**) presents the average duration of an exposure event in a district over time.

## Defining heat stress exposure (HSE)

We define HSE as the number of hours in the day that UTCI crosses a threshold of 32 °C. Details are provided in Eq. (3) and Eq. (4).

$$HSE_d = \sum_{t=1}^{24} HSE_h \qquad (3)$$

$$HSE_h = 1, \text{if } UTCI_h > 32°C; HSE_h = 0, \text{otherwise} \qquad (4)$$

where $HSE_d$ is the exposure for a given day, and $HSE_h$ is the exposure for a given hour. $HSE_h$ equals 1 if $UTCI_h$ (which is the UTCI for that hour) exceeds 32 °C, and $HSE_h$ equals 0 otherwise.

## District geometry

Our analysis incorporates 722 districts in India. The updated district boundaries are derived from prior work done by Jain et al. (2024)[40]. These boundaries include the recent district changes in the state of Andhra Pradesh in India, in which 13 new districts were created.

## Robust trend estimates

Figure 2f plots the trend in HSE (Y-axis) against the average daily HSE (X-axis) for all districts in India. To calculate the trend in HSE, we use robust Sen's slope estimates (also known as the Theil Sen estimator)[41,42]. Theil (1950) proposed the median of the pairwise slopes as an estimator of the slope for a simple linear model[41]. Sen (1968) extended this to resolve ties[42]. The Theil-Sen estimator has been widely used to compute trends in climate data[43,44]. To compute the trend in exposure, we first calculate the monthly time series of exposure for each district from 1981 to 2023. The mean daily exposure experienced during a month is the simple average of the total number of hours of exposure for each day in the month. The Theil-Sen estimator is then computed using the TheilSen function in the openair package in R, which allows the trend estimate to be calculated after accounting for seasonality[45]. The analysis for Panel B of Fig. 4 also follows a similar approach.

## Outdoor occupational data

To estimate outdoor occupational exposure in Fig. 7, we use data from the Consumer Pyramids Household Survey (CPHS). The CPHS is a nationally representative panel survey that collects information from approximately 170,000 households in successive waves, with each wave occurring once every four months. Although the CPHS began in January 2014, the survey first added questions on time spent at work and the location of work in the wave that started on September 1, 2019. For all individuals reporting non-zero work hours, we classify them as *outdoor* workers if their place of work is "Own farm," "Other's farm," "Bhagidari/Leased farm," or "Marketplace"; all others are classified as *indoor* workers.

The CPHS identifies the district of residence for each individual, which we use to merge the survey with district-level UTCI data. We average the UTCI-based heat-stress exposure for the duration of each wave at the district level. An individual's exposure is defined as per Eq. (5)[11].

$$Outdoor\ exposure_{ijw} = \begin{cases} HSE_{iw}, \text{if } OWH_{iw} \geq HSE_{jw} \\ OWH_{jw}, \text{if } HSE_{jw} > OWH_{iw} \end{cases} \qquad (5)$$

Where $Outdoor\ exposure_{ijw}$ is the outdoor occupational exposure of individual $i$ in district $j$ in wave $w$, $OWH_{iw}$ is the outdoor workhours of individual $i$ in wave $w$, and $HSE_{jw}$ is the average heat stress exposure experienced by district $j$ during wave $w$.

For Fig. 7a we sum the individual exposures (after applying household weights and adjusting for missing responses) to obtain the aggregate exposure in each wave and compute the proportion of outdoor work hours that are exposed (OutdoorExposure / OWH) (*Outdoor exposure*/*OWH*).

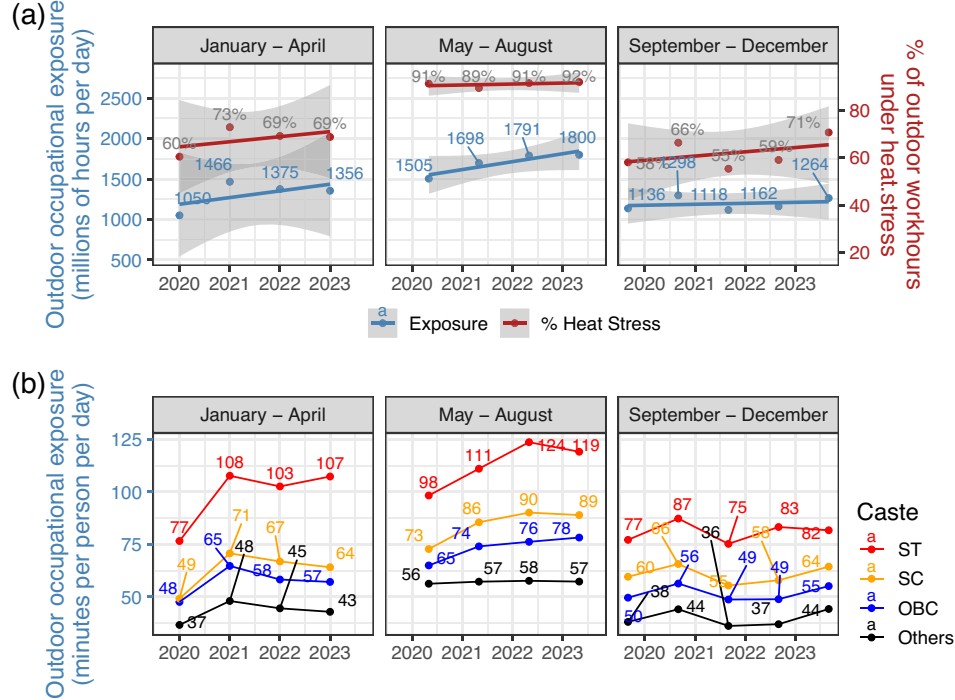

**Fig. 7 | Outdoor occupational exposure from 2019 to 2023.** The figure presents changes in outdoor occupational exposure from September 2019 to December 2023. Outdoor exposure occurs when an individual is working outdoors at a time when the Universal Thermal Climate Index (UTCI)' exceeds 32 °C. Panel (**a**) presents changes in aggregate outdoor occupational exposure across India (left Y-axis) and the proportion of outdoor work hours that are exposed (right Y-axis). The line is the OLS regression for the set of points in the graph. The shaded region represents the 95% confidence interval for the regression line. Panel (**b**) presents the average outdoor occupational exposure for individuals from different caste groups.

For Fig. 7b we compute average individual outdoor exposure separately for caste groups, using the caste identity recorded in the survey. Scheduled Castes (SC) are historically "untouchable" communities that remain marginalized in modern India. Scheduled Tribes (ST) are tribal communities that have historically lived in relative isolation. Other Backward Classes (OBC) are socially marginalized but politically influential groups. "Others" is the residual category and can be considered socially and politically dominant[46]. It is important to note that SC and ST are official government categories in India.

### Reporting summary
Further information on research design is available in the Nature Portfolio Reporting Summary linked to this article.

## Data availability
The data and code for replicating the figures are available here: https://doi.org/10.6084/m9.figshare.29501147.

## Code availability
The data and code for replicating the figures are available here: https://doi.org/10.6084/m9.figshare.29501147.

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

## Acknowledgments

This research was funded by the Bill & Melinda Gates Foundation, INV-002992 (RK, SVS). The funder had no role in design and conduct of the study; collection, management, analysis, and interpretation of the data; preparation, review, or approval of the manuscript; and decision to submit the manuscript for publication.

## Author contributions

All authors conceptualized the study. S.V.S. supervised the study. A. Shah led the analysis and the writing of the draft. A. Sugathan, D.M., R.K. and S.V.S. contributed to the data interpretation and critical review & editing of the draft. All authors had final responsibility for the decision to submit for publication.

## Competing interests

The authors declare no competing interests.
