## [Transparent Peer Review file · Nature Communications]

SPATIOTEMPORAL CHANGES IN HEAT STRESS EXPOSURE IN INDIA, 1981-2023

Corresponding Author: Professor S V Subramanian

Version 0:

Reviewer comments:

Reviewer #1

(Remarks to the Author)

Review for Spatiotemporal changes in heat stress exposure in India, 1981-2023, by Arpit Shah et al. (2024).

This manuscript examines the Spatiotemporal changes in heat stress exposure in India, 1981-2023, using various land surface datasets. Extreme climate events continue to pose threats and devastating impacts to society, economy, and ecosystem across many regions globally. A comprehensive understanding of their development and occurrences remains relevant, with key mechanisms initiating and amplifying them. From this point of view, examining economic development factors associated with the recent occurrence of extreme heatwaves over vulnerable regions is of great importance, as well as information pertinent to the scientific community. Overall, the study is interesting and well-written. However, significant flaws exist in the present research approach and the results reported in the current manuscript. Therefore, the reviewer would like to recommend that this manuscript be returned to the authors for major revisions, as suggested below, before a revised manuscript may be re-submitted for publication. My comments are given below;

Overall, the methods do not use the land-surface datasets capacity for higher precision heat-related estimates of population changes. The techniques utilized in this manuscript were a standard in previous work due to the difficulty quantifying temperature-radiation-humidity relationships. However, recent work over the past few years has dramatically improved the capacity to quantify the socio-economic impacts of climate change. Numerous code modules are available, and various institution's datasets have sub-daily variables to calculate the enhanced metrics. These datasets bring high-resolution and sub-daily covariances of temperature, humidity, and radiation for global-scale models. This enables sophisticated measures to produce heat stress-related exposures. Recently, many efforts have incorporated methods for quantifying labourers' exposure to heat utilizing global output.

Furthermore, these new methods are grounded in the strengths of land-surface models balance and moist thermodynamics-which produce far superior results that will stand up to scrutiny (Buzan and Huber, 2020; McKinnon and Poppick, 2020; Poppick and McKinnon, 2020; Schwingshackl et al., 2021). I recommend that the authors consider utilizing these new methods, i.e. Environmental Stress Index or Wet Bulb Globe Temperature, along with labour capacity functions, such as the NIOSH labour standards (de Lima et al., 2021; Orlov et al., 2021). Also, these datasets diverge from each other in time, however, their per degree of global warming change is robust. Presenting the results in terms of per degree of warming instead of in time would reduce the error bars in the projects, limiting the errors to the divergence in estimated population rather than the combined error of the amount of climate change and population changes. Some pitfalls to avoid: use the same temporal frequency covering variables. The underlying algorithms are non-linear, and Reynolds averaging (Buzan et al., 2015) demonstrates that the errors can be on the scale of climate change if not appropriately treated (Buzan and Huber, 2020). Additionally, if WBGT is used, avoid using Stull 2011 as the wet bulb temperature calculation. The function was not designed for use outside of 25{degree sign}C Tw, and errors can be considerably significant, inflating heat stress exposure. Recommended is to use Davies-Jones 2008. <http://dx.doi.org/10.1038/s41612-024-00764-5>

<https://agupubs.onlinelibrary.wiley.com/doi/10.1029/2021EF002240>

Besides, both equations 1 and 2 were used for HSE for exposure; however, it is unclear which one is utilised for the final output. Both equations contradict with fewer explanations.

In the Introduction: The authors should point out this manuscript's research goal and emphasise its significance. Moreover, the authors should introduce the implications of previous related published research from a broader perspective.

What's the principle of the threshold of heatwave long-duration events selected? The threshold of the heatwave to identify the duration seems unreasonable and explainable.

I feel the authors need to discuss the study's limitations in the discussion section and what needs to be done to address those. For example, a comprehensive analysis of climate sensitivity, given the datasets (that are used in the study), has a

significant uncertainty amongst their models. Although I see that the authors have mentioned this in their manuscript, it still needs more elaborate discussion. Finally, discuss some lights on the physical mechanism on which the extreme heat wave is increasing over the regions with the help of some recent literature.

<https://onlinelibrary.wiley.com/doi/10.1029/2021EF002240>

Conclusions are not clarified. Besides, the data and results cannot support these findings well.

(Remarks on code availability)

None

Reviewer #2

(Remarks to the Author)

This is a very clearly written manuscript about the spatial and temporal patterns of UTCI and HSE across India. The results as well as the methodology are clearly written and there is really not much to comment. Although the study brings interesting and quite important findings for the region, I am not sure if it is "ground-breaking" enough to be published in Nature Comm. But that is rather up to the Editors than me.

(Remarks on code availability)

I did not find any code attached.

Version 1:

Reviewer comments:

Reviewer #1

(Remarks to the Author)

2nd Review for Spatiotemporal changes in heat stress exposure in India, 1981-2023, by Arpit Shah et al. (2024).

The authors went through a great length of improvement and revised the manuscript in general while avoiding novelty and further articulation. However, significant improvements exist in the present research approach, and the results are reported in the current manuscript. Therefore, the reviewer would like to recommend that this manuscript be returned to the authors for minor revisions, as suggested below, before a revised manuscript may be re-submitted for publication. My comments are given below;

-The 32°C UTCI threshold for HSE lacks physiological or epidemiological validation. While cited as "stressful for outdoor work," no references link this threshold to Indian occupational contexts (e.g., agricultural/construction workers). Provide empirical evidence (e.g., from de Lima et al., 2021) or adjust thresholds based on activity-specific metabolic rates.

<https://agupubs.onlinelibrary.wiley.com/doi/10.1029/2021EF002240>

-Equations 1 (UTCI) and 2–3 (HSE) are now clearer, but the polynomial function for UTCI (mentioned in Methods) is not explicitly provided. The sixth-order polynomial cited by Bröde et al. (2012) is non-linear; it ensures computational reproducibility by including the full equation or a reference to open-source code.

-While ERA5/ERA5-Land are state-of-the-art, the authors do not leverage newer datasets (e.g., HIGTS, Jian et al., 2024) or sub-daily covariances to improve precision. The manuscript claims to use "high temporal resolution" but does not exploit hourly data to model intra-day heat stress dynamics (e.g., recovery periods between heat episodes).

-The primary novelty—long-term (1981–2023), high-resolution (0.1°, hourly) HSE analysis—is commendable but not unique. Recent studies (e.g., HIGTS, Yan et al., 2021) have achieved similar spatiotemporal granularity. The authors must explicitly contrast their work with these datasets, emphasizing improvements (e.g., district-level governance focus, longer temporal coverage).

-The district-level focus is policy-relevant but underdeveloped. Link HSE trends to district-specific vulnerabilities (e.g., caste-based occupational disparities, healthcare access) using socioeconomic data from Subramanian et al. (2023). Currently, the analysis remains descriptive rather than actionable.

-While dataset biases (ERA5/ERA5-Land) are noted, the discussion does not address systemic limitations of UTCI, such as its assumption of "reference clothing" or metabolic rates irrelevant to Indian labourers. Compare UTCI-derived HSE with WBGT-based estimates to quantify methodological uncertainties.

(Remarks on code availability)

Please share the codes.

Version 2:

Reviewer comments:

Reviewer #1

(Remarks to the Author)

I have reviewed the revised manuscript and the author's point-by-point response. I am pleased to find that the authors have done an excellent job addressing all the concerns I raised. I have no further requests for revision and recommend acceptance of the manuscript in its current form.

(Remarks on code availability)

None

Response to Reviewers for “Spatiotemporal changes in heat stress exposure in India, 1981-2023”
Revision 1

Reviewer 1

We want to thank Reviewer 1 for their detailed comments and for the helpful references they have provided us.

Comment	Response
This manuscript examines the Spatiotemporal changes in heat stress exposure in India, 1981-2023, using various land surface datasets. Extreme climate events continue to pose threats and devastating impacts to society, economy, and ecosystem across many regions globally. A comprehensive understanding of their development and occurrences remains relevant, with key mechanisms initiating and amplifying them.	Thank you for this comment. We agree with Reviewer 1 that comprehensive examination of the development and evolution of extreme climatic events such as heat is highly relevant today. Our work takes a step in that direction by analyzing the spatiotemporal evolution of UTCI from 1981 to 2023 at the district level in India. Since districts serve as a key site for health-related policymaking in India, our analysis provides data at a scale directly relevant to policy decisions.
From this point of view, examining economic development factors associated with the recent occurrence of extreme heatwaves over vulnerable regions is of great importance, as well as information pertinent to the scientific community.	Thank you for this comment. We agree that integrating data on economic development over vulnerable regions with episodes of excess heat is important both from an economic and a health perspective. Research suggests that outdoor workers in physically laborious occupations such as construction work and agriculture can be highly vulnerable to excess heat (Shah et al., 2025). In addition, indoor workers in poorly ventilated or crowded settings such as factories or garment manufacturing units could also face significant heat burdens. In a different paper, we use data from 2019 and 2022 to argue that caste is an important determinant of the outdoor heat exposure of workers in India (Shah et al., 2025). In this paper, we focus on changes in heat stress exposure in India’s districts from 1981 to 2023. While our focus here is restricted to the evolution of heat, our work on how district-level UTCI in India has changed from

	1981 to 2023 can provide a starting point for researchers attempting to understand relationships between the economy and heat exposure. Future work building on this research can examine vulnerable regions, inequality in heat exposure, and the health impacts of heat exposure, among other possibilities.
Overall, the study is interesting and well-written. However, significant flaws exist in the present research approach and the results reported in the current manuscript. Therefore, the reviewer would like to recommend that this manuscript be returned to the authors for major revisions, as suggested below, before a revised manuscript may be re-submitted for publication.	Thank you for the constructive feedback on the manuscript and for the opportunity to revise this work. We appreciate Reviewer 1's expertise that has gone into the comments. We are also appreciative of the opportunity to revise the manuscript.
Overall, the methods do not use the land-surface datasets capacity for higher precision heat-related estimates of population changes. The techniques utilized in this manuscript were a standard in previous work due to the difficulty quantifying temperature-radiation-humidity relationships. However, recent work over the past few years has dramatically improved the capacity to quantify the socio-economic impacts of climate change. Numerous code modules are available, and various institution's datasets have sub-daily variables to calculate the enhanced metrics. These datasets bring high-resolution and sub-daily covariances of temperature, humidity, and radiation for global-scale models. This enables sophisticated measures to produce heat stress-related exposures.	We would like to thank Reviewer 1 for this detailed comment. Our main objective is to assess the evolution of heat stress in India from 1981 to 2023 at a scale that is relevant for policymaking. Given that districts play a central role in health governance (Subramanian et al., 2023), we employ the district as the unit of analysis. The average Indian district spans ~4,500 square kilometres. Our UTCI data, which we aggregate to a district level, has a resolution of 0.1° (average pixel size of 120 square km). This implies that the average district comprises 40+ UTCI readings, which is suitably granular for our analysis. Higher spatial resolution would not necessarily improve our accuracy, as our focus remains on aggregated district-level assessments. Given our research objective of doing a district-level analysis, we prioritized a metric that has data availability at a high temporal resolution, and is considered a strong proxy for the wet bulb globe temperature. The Universal Thermal Climate Index (UTCI) fits the bill perfectly for our purpose as data is available over a long period at an hourly temporal resolution. The UTCI was

	developed by the International Society of Biometeorology to overcome shortcomings in earlier two-parameter heat indices (that used only air temperature and proxies for humidity). The UTCI incorporates air temperature, humidity, solar radiation and wind speed and is designed to be appropriate for assessments of outdoor thermal conditions in human biometeorological applications (including epidemiological research) (Jendritzky et al., 2012). The creation of high-resolution UTCI metrics represents a recent advance in the literature on heat that we leverage for this paper. As the reviewer has also suggested, we develop the UTCI data at a district level by adapting a recently developed code module to fit our purpose (Yan et al., 2021).
Recently, many efforts have incorporated methods for quantifying labourers' exposure to heat utilizing global output.	Thank you for this comment. We agree with Reviewer 1 that quantifying laborer's exposure to heat is extremely important from an economic and health perspective. While our focus here is restricted to a spatiotemporal analysis of changes in UTCI and heat stress exposure, we are confident that this paper offers a great starting point for future research in this space.
Furthermore, these new methods are grounded in the strengths of land-surface models balance and moist thermodynamics-which produce far superior results that will stand up to scrutiny (Buzan and Huber, 2020; McKinnon and Poppick, 2020; Poppick and McKinnon, 2020; Schwingshackl et al., 2021).	We provide a brief review of the excellent papers that Reviewer 1 has cited, and how they apply to our research question. Buzan and Huber (2020) focus on developing a framework to enable robust predictions of moist heat stress and its spatial distribution in the future. They provide an comprehensive review of the climatological science and modelling approaches relevant for future projects of moist heat stress metrics. McKinnon and Poppick (2020) develop a semi-parametric model to assess how the dew point changes with near-surface temperature under conditions of increasing

	global mean temperatures. The objective is to allow combined assessments of heat-humidity risks and inform future climate model projections that attempt to predict moist heat stress. Poppick and McKinnon (2020) leverage their approach to simulate future temperature and humidity. They combine real-world weather station observations with projections from global climate models to inform future simulations of temperature and humidity over the continental United States. Schwingshackl et al. (2021) compute projections of eight heat stress indices (including the UTCI) as a function of future global mean temperature. They find projected increases in heat across all metrics. They stress the need to choose metrics based on the application/research question. In contrast to Buzan and Huber (2020), McKinnon and Poppick (2020), Poppick and McKinnon (2020), and Schwingshackl et al. (2021), our work is primarily focused on a historical analysis of district-level heat stress and we are not attempting future projections of UTCI and heat stress exposure. In addition, most of the cited papers aggregate sub daily data (usually readings every 4 to 6 hours) into daily data for analysis. The typical spatial resolution is 0.25°. These datasets are appropriate for the research questions that the papers are interested in. In contrast, we are interested in a higher resolution portrait of heat stress evolution in India's districts. As we discuss earlier, 0.1° spatial resolution is appropriate for our needs. In addition, an hourly temporal resolution enables more precise estimates of daily heat stress levels. Prior work on heat waves typically assumes heat waves to occur if heat crosses certain thresholds at any point during the day. By capturing the number of hours for which heat actually crosses threshold values on a
--	---

	daily basis, we are better able to estimate heat stress exposure.
I recommend that the authors consider utilizing these new methods, i.e. Environmental Stress Index or Wet Bulb Globe Temperature, ...	Thank you for this comment. The Environmental Stress Index, developed in 2001, considers air temperature, relative humidity and solar radiation to assess thermal stress and was designed as a proxy for the WBGT (Moran et al., 2001). The UTCI, developed in 2012, can be considered as advancement on this front. As we indicate, the UTCI is a highly reliable proxy for the WBGT under a variety of conditions and is suitable for heat stress assessments (Blazejczyk et al., 2012). The UTCI is also mentioned as one of the heat metrics in two of the papers referenced by Reviewer 1 in an earlier comment (Buzan et al., 2015; Schwingshackl et al., 2021).
...along with labour capacity functions, such as the NIOSH labour standards (de Lima et al., 2021; Orlov et al., 2021).	Thank you for this comment. de Lima et al. (2021) consider the impact on output of crops under increasingly hot climatic conditions. Their analysis considers two channels through which crop output can be impacted – the first is decreased labor productivity because of heat stress (calculated using labor response functions), and the second is the direct impact of temperature on the plant yields. They find that both labor and yield impacts are equally important for projected global warming of 3°C. Similar to de Lima et al. (2021), Orlov et al. (2021) calculate crop loss as a combined effect of the impact of heat on labor productivity and crop yields. They find an overall negative impact, with the decline in labor productivity offsetting any potential gains from increase carbon dioxide fertilization at higher temperatures. While de Lima et al. (2021) and Orlov et al. (2021) are providing future projections of economic impacts using labor loss functions

	for workers in a particular sector (agriculture) and for particular crops, our analysis is restricted to the historical evolution of heat stress in India's districts.
Also, the these datasets diverge from each other in time, however, their per degree of global warming change is robust. Presenting the results in terms of per degree of warming instead of in time would reduce the error bars in the projects, limiting the errors to the divergence in estimated population rather than the combined error of the amount of climate change and population changes. Some pitfalls to avoid: use the same temporal frequency covering variables. The underlying algorithms are non-linear, and Reynolds averaging (Buzan et al., 2015) demonstrates that the errors can be on the scale of climate change if not appropriately treated (Buzan and Huber, 2020). Additionally, if WBGT is used, avoid using Stull 2011 as the wet bulb temperature calculation. The function was not designed for use outside of 25{degree sign}C Tw, and errors can be considerably significant, inflating heat stress exposure. Recommended is to use Davies-Jones 2008; IrfanUllah 2022.	Thank you for this comment. Our analysis is restricted to estimating heat stress evolution over India's districts from 1981 to 2023 – we are not considering population changes in our approach. Our research can be considered a starting point for future research to build upon in terms of assessing the impacts of heat on economic activity and health. As Reviewer 1 rightly points out, future research would need to take into account the combined impacts of climate change and population changes. Since our work is focused only on historical data and not on future projections, estimation errors are restricted to the typical error range involved in historical ERA datasets (Muñoz-Sabater et al., 2021). Regarding the pitfalls that Reviewer 1 rightly points out, we compute the UTCI using ERA5 and ERA5-Land data on variables with the same temporal and spatial resolution (Yan et al., 2021). Accordingly, we do not face issues involved with geographical and temporal aggregation and/or downscaling. Our analysis leverages a standard and widely accepted approach to compute the UTCI (Yan et al., 2021), which in turn is based on Bröde et al. (2012).
Besides, both equations 1 and 2 were used for HSE for exposure; however, it is unclear which one is utilised for the final output. Both equations contradict with fewer explanations. What's the principle of the threshold of heatwave long-duration events selected? The threshold of the heatwave to identify the duration seems unreasonable and explainable.	Equation 1 in the paper is the formula used to compute the UTCI. This is based on the widely accepted approach developed by Bröde et al. (2012). Equations 2 and 3 provide the details for computing heat stress exposure (HSE) from the UTCI values. We define HSE as occurring when UTCI crosses 32°C in a given hour (Equation 3). The thresholds are based on work done by Błażejczyk et al. (2013). They characterize UTCI exceeding 32°C as 'strong heat stress' conditions where the human physiological

	response includes higher levels of sweating, instantaneous changes in skin temperature, higher rectal temperatures, and higher latent heat loss (page 8, Błażejczyk et al., 2013). The total HSE during a day is the total number of hours for which UTCI exceeds 32°C in the day (Equation 2). Our analysis is based on the evolution of HSE at the district level in India. This approach yields more precise estimates as compared to prior work that assumes a heat wave for the day if heat crosses a certain threshold at any point during the day. Using hourly information also enables certain types of analysis – such as our analysis of the diurnal range and the analysis of HSE at different times of day.
In the Introduction: The authors should point out this manuscript's research goal and emphasise its significance. Moreover, the authors should introduce the implications of previous related published research from a broader perspective.	Thank you for this comment. As suggested by Reviewer 1, we have updated the introduction to highlight the research objective and significance of our work. We aim to provide a detailed portrait of the historical evolution of heat stress exposure in India from 1981 to 2023 at a policy relevant scale. We select the district as the unit of analysis because of its importance as a site for policy making in India's federal structure. Our analysis leverages high-resolution (0.1°, hourly) data on the Universal Thermal Climate Index (UTCI). We measure heat stress exposure (HSE) as the number of hours in a day that UTCI crosses a threshold of 32°C, a level that can be considered highly stressful for human beings. In doing so, we go beyond prior work that has examined the evolution of heat stress in India (or models that have made future predictions of heat stress under different climate change scenarios) – these efforts have typically relied on daily data (in some cases, sub-daily data at a six-hourly level) with a spatial resolution of 0.25°. We would like to thank Reviewer 1 for the extensive citations they provided in their comments – these references have helped us frame and improve our argument.

I feel the authors need to discuss the study's limitations in the discussion section and what needs to be done to address those. For example, a comprehensive analysis of climate sensitivity, given the datasets (that are used in the study), has a significant uncertainty amongst their models. Although I see that the authors have mentioned this in their manuscript, it still needs more elaborate discussion.	Thank you for this comment. We have provided a discussion on the limitations of the UTCI calculations in the Discussion section. While a complete discussion on potential biases in UTCI calculations is beyond the scope of this manuscript, we refer readers to the original manuscript that we draw upon with regard to our computational methods (Yan et al., 2021). Currently, hourly thermal information that can enable us to compute moist heat stress is available at a spatial resolution of 0.1°. If such information were available at higher resolutions (say 30 m), it would be possible to assess the role of factors such the urban heat island effect is influencing heat stress. Current data limitations make this difficult. For instance, Landsat data is available at 30 m, but Landsat only provides 1 reading every fortnight.
Finally, discuss some lights on the physical mechanism on which the extreme heat wave is increasing over the regions with the help of some recent literature.	Thank you for this comment. Scientists have generally argued for two reasons for the increase in moist heat stress in India. The first reason is global warming, which has caused an increase in heat in many parts of the world (Buzan and Huber, 2020). The second reason, which is particular to India, is the increased prevalence of irrigation around the country during the same time frame of our study. Widespread irrigation can lead to increased humidity and higher moist heat stress (Mishra et al., 2020).
Conclusions are not clarified. Besides, the data and results cannot support these findings well.	Thank you for this comment. Through the changes made in the Introduction and Discussion sections, we have attempted to ensure that the research objectives and significance are clearly highlighted. We hope that these changes also help clarify how our data and methodological approach help support the conclusions of the manuscript.

Reviewer 2

We want to thank Reviewer 2 for their encouraging comments on the writing and methodological details of the manuscript.

Comment	Response
This is a very clearly written manuscript about the spatial and temporal patterns of UTCI and HSE across India. The results as well as the methodology are clearly written and there is really not much to comment.	Thank you for the encouraging comment.
Although the study brings interesting and quite important findings for the region, I am not sure if it is "ground-breaking" enough to be published in Nature Comm. But that is rather up to the Editors than me.	Our work presents the most detailed and comprehensive assessment of long-term heat stress exposure in India at spatial scales that are policy relevant. We leverage an improved thermal metric, namely, the Universal Thermal Climate Index, which combines air temperature, humidity, solar radiation and wind speed to provide a better understanding of heat stress as compared to older approaches that relied on air temperature. Our temporal coverage (1981-2023) also goes beyond existing efforts. Prior work on understanding heat has largely focused on defining entire days as 'heat-wave' depending on whether temperatures cross certain thresholds. We use hourly information to show that there are intra-day increases over time in the number of hours/day where UTCI is crossing thresholds that can be considered stressful for human beings.

References

- Blazejczyk, Krzysztof, et al. "Comparison of UTCI to selected thermal indices." *International journal of biometeorology* 56 (2012): 515-535.
- Błażejczyk, Krzysztof, et al. "An introduction to the universal thermal climate index (UTCI)." *Geographia Polonica* 86.1 (2013): 5-10.
- Buzan, J. R., Keith Oleson, and Matthew Huber. "Implementation and comparison of a suite of heat stress metrics within the Community Land Model version 4.5." *Geoscientific Model Development* 8.2 (2015): 151-170.
- Buzan, Jonathan R., and Matthew Huber. "Moist heat stress on a hotter Earth." *Annual Review of Earth and Planetary Sciences* 48.1 (2020): 623-655.
- Bröde, Peter, et al. "Deriving the operational procedure for the Universal Thermal Climate Index (UTCI)." *International journal of biometeorology* 56 (2012): 481-494.
- Buzan, Jonathan R., and Matthew Huber. "Moist heat stress on a hotter Earth." *Annual Review of Earth and Planetary Sciences* 48.1 (2020): 623-655.
- Davies-Jones, Robert. "An efficient and accurate method for computing the wet-bulb temperature along pseudoadiabats." *Monthly Weather Review* 136.7 (2008): 2764-2785.
- De Lima, Cicero Z., et al. "Heat stress on agricultural workers exacerbates crop impacts of climate change." *Environmental Research Letters* 16.4 (2021): 044020.
- Ullah, Irfan, et al. "Projected changes in socioeconomic exposure to heatwaves in South Asia under changing climate." *Earth's Future* 10.2 (2022): e2021EF002240.
- Jendritzky, Gerd, Richard De Dear, and George Havenith. "UTCI—why another thermal index?." *International journal of biometeorology* 56 (2012): 421-428.
- McKinnon, Karen A., and Andrew Poppick. "Estimating changes in the observed relationship between humidity and temperature using noncrossing quantile smoothing splines." *Journal of Agricultural, Biological and Environmental Statistics* 25 (2020): 292-314.
- Mishra, Vimal, et al. "Moist heat stress extremes in India enhanced by irrigation." *Nature Geoscience* 13.11 (2020): 722-728.
- Moran, Daniel S., et al. "An environmental stress index (ESI) as a substitute for the wet bulb globe temperature (WBGT)." *Journal of thermal biology* 26.4-5 (2001): 427-431.
- Muñoz-Sabater, Joaquín, et al. "ERA5-Land: A state-of-the-art global reanalysis dataset for land applications." *Earth system science data* 13.9 (2021): 4349-4383.
- Orlov, Anton, et al. "Global economic responses to heat stress impacts on worker productivity in crop production." *Economics of Disasters and Climate Change* 5 (2021): 367-390.
- Poppick, Andrew, and Karen A. McKinnon. "Observation-based simulations of humidity and temperature using quantile regression." *Journal of Climate* 33.24 (2020): 10691-10706.
- Shah, Arpit, et al. "Caste inequality in occupational exposure to heat waves in India." *Demography* 62.1 (2025): 35-60.

Subramanian, S. V., et al. "Progress on Sustainable Development Goal indicators in 707 districts of India: a quantitative mid-line assessment using the National Family Health Surveys, 2016 and 2021." *The Lancet Regional Health-Southeast Asia* 13 (2023).

Schwingshackl, Clemens, et al. "Heat stress indicators in CMIP6: Estimating future trends and exceedances of impact-relevant thresholds." *Earth's Future* 9.3 (2021): e2020EF001885.

Yan, Y., Xu, Y. & Yue, S. A high-spatial-resolution dataset of human thermal stress indices over South and East Asia. *Scientific data* 8, 229 (2021)

Response to Reviewers for "Spatiotemporal changes in heat stress exposure in India, 1981-2023"

Revision 2

We thank the Handling Editor and the review team for their constructive comments that have helped us shape and improve the manuscript. We are grateful for their time and effort.

Reviewer 1

Thank you for your excellent comments on the paper.

Comment	Response
The authors went through a great length of improvement and revised the manuscript in general while avoiding novelty and further articulation. However, significant improvements exist in the present research approach, and the results are reported in the current manuscript. Therefore, the reviewer would like to recommend that this manuscript be returned to the authors for minor revisions, as suggested below, before a revised manuscript may be re-submitted for publication. My comments are given below;	Thank you for this comment and the encouragement and direction you provided during the review process.
-The 32°C UTCI threshold for HSE lacks physiological or epidemiological validation. While cited as "stressful for outdoor work," no references link this threshold to Indian occupational contexts (e.g., agricultural/construction workers). Provide empirical evidence (e.g., from de Lima et al., 2021) or adjust thresholds based on activity-specific metabolic rates. https://agupubs.onlinelibrary.wiley.com/doi/10.1029/2021EF002240	The UTCI is based on an assumption of reference clothing and activity level (2.3 MET or 135 W/m², equivalent to walking 4 km/hour)¹. The reviewer is correct in pointing out that thresholds will vary based on activity-specific metabolic rates. While we have not systematically analyzed activity-specific thresholds for this work, it remains an important area for future research. Our usage of the 32°C UTCI threshold is based on physiological validation done by Błażejczyk et al. (2013)¹. The 32°C threshold indicates intense heat stress under reference clothing and activity conditions.

	While we do not adjust thresholds for specific activities, it is important to note that our methodology is flexible and can be adapted for sector-specific thresholds in a straightforward manner. Given that many workers in outdoor jobs are doing manual labor with higher than reference activity levels, it is possible that our analysis underestimates heat stress exposure for the most vulnerable parts of the labor force (Figure 7 in the revised manuscript uses the 32°C to estimate outdoor heat stress exposure using labor force survey data – details are provided in the Revised Manuscript).
-Equations 1 (UTCI) and 2–3 (HSE) are now clearer, but the polynomial function for UTCI (mentioned in Methods) is not explicitly provided. The sixth-order polynomial cited by Bröde et al. (2012) is non-linear; it ensures computational reproducibility by including the full equation or a reference to open-source code.	Thank you for this comment. We have provided a more detailed version of the formula and included references to the original polynomial form used for the UTCI computation² and source code³.
-While ERA5/ERA5-Land are state-of-the-art, the authors do not leverage newer datasets (e.g., HIGTS, Jian et al., 2024) or sub-daily covariances to improve precision. The manuscript claims to use "high temporal resolution" but does not exploit hourly data to model intra-day heat stress dynamics (e.g., recovery periods between heat episodes).	Thank you for highlighting this. We provide a brief comparison of our work with the HIGTS and HiTiSea datasets in the paper – our work has a larger temporal span and/or better spatial/temporal resolution than these datasets. To more effectively address the reviewer’s comment, we have included additional analyses that leverage our hourly data. To do so, we define a heat stress exposure event as a continuous period during which UTCI exceeds 32°C in a given location. Figure 6 in the revised manuscript examines how the number of such events, their duration, and the interval between them have changed over time. Details are provided in the revised manuscript.
-The primary novelty—long-term (1981–2023), high-resolution (0.1°, hourly) HSE analysis—is commendable but not unique. Recent studies (e.g., HIGTS, Yan	Thank you for pointing this out. We have updated the text to highlight the novelty of our work.

et al., 2021) have achieved similar spatiotemporal granularity. The authors must explicitly contrast their work with these datasets, emphasizing improvements (e.g., district-level governance focus, longer temporal coverage).	
-The district-level focus is policy-relevant but underdeveloped. Link HSE trends to district-specific vulnerabilities (e.g., caste-based occupational disparities, healthcare access) using socioeconomic data from Subramanian et al. (2023). Currently, the analysis remains descriptive rather than actionable.	Thank you for this important comment. In this revision, we have linked heat stress exposure at the district to outdoor occupational exposure using labor force data. Specifically, we match occupational information from consecutive waves of the Consumer Pyramids Household Survey (CPHS) from September 2019 to December 2023 with heat stress exposure information based on the UTCI data. The CPHS allows us to estimate outdoor occupational exposure data based on the location where the individual is working. We also estimate outdoor occupational exposure for different caste groups based on the same data. We have included details regarding the methodology and results in the Revised Manuscript.
-While dataset biases (ERA5/ERA5-Land) are noted, the discussion does not address systemic limitations of UTCI, such as its assumption of reference clothing or metabolic rates irrelevant to Indian labourers. Compare UTCI-derived HSE with WBGT-based estimates to quantify methodological uncertainties.	Thank you for this comment. We agree that the UTCI metric has limitations. Li et al. (2020)⁴ make publicly available a global WBGT dataset (calculated using the simplified WBGT approach) at a resolution of 0.25°. Using our approach, we extract hourly WBGT values for 2020-2022 from this dataset and compare it to hourly UTCI estimates at the district level. The correlation between WBGT and UTCI is 0.85, indicating the suitability of the UTCI as a heat stress metric. When we compare daily average WBGT and UTCI values at the district level for 2020-22, the correlation increases to 0.96. In addition, we would also like to highlight other research that has examined the reliability of the UTCI as a metric for heat stress. First, recent work by Havenith et al (2024)⁵ compared the efficacy of different thermal

	indices, including the UTCI and WBGT, for predictions of labor loss, body temperatures and thermal perception in a wide variety of warm and hot climates. Results indicates that the UTCI performed as well as the WBGT in predicting human physiological responses (and better than the WBGT in some cases). The UTCI also out-performed all other indicators that were considered in the study. Second, the UTCI is based on a multi-node model of human heat transfer and thermo-regulation and has been recommended by researchers as an alternative to the WBGT, especially in conditions where individuals are doing high intensity activities⁶.
Reviewer #1 (Remarks on code availability): Please share the codes.	Please find a link to the codes here: https://figshare.com/s/576ef1f0723e053755fd

References

1. Blazejczyk, K., Epstein, Y., Jendritzky, G., Staiger, H. & Tinz, B. Comparison of UTCI to selected thermal indices. *Int J Biometeorol* **56**, 515–535 (2012).
2. Bröde, P. *et al.* Deriving the operational procedure for the Universal Thermal Climate Index (UTCI). *International journal of biometeorology* **56**, 481–494 (2012).
3. Yan, Y., Xu, Y. & Yue, S. A high-spatial-resolution dataset of human thermal stress indices over South and East Asia. *Scientific data* **8**, 229 (2021).
4. Li, D., Yuan, J. & Kopp, R. E. Escalating global exposure to compound heat-humidity extremes with warming. *Environ. Res. Lett.* **15**, 064003 (2020).
5. Havenith, G., Smallcombe, J. W., Hodder, S., Jay, O. & Foster, J. Comparing the efficacy of different climate indices for prediction of labor loss, body temperatures, and thermal perception in a wide variety of warm and hot climates. *Journal of Applied Physiology* **137**, 312–328 (2024).
6. Błażejczyk, K. *et al.* An introduction to the Universal Thermal Climate Index (UTCI). *Geogr. Pol.* **86**, 5–10 (2013).